# Randomised controlled trial of a behaviour change physiotherapy intervention to increase physical activity following hip and knee replacement: the PEP-TALK trial

Toby O Smith,[1,2] Scott Parsons,[1] Alexander Ooms,[1,3] Susan Dutton,[1,3] Beth Fordham  ,[1] Angela Garrett,[1] Caroline Hing,[4] Sarah Lamb,[5] on behalf of the PEP-TALK Trial Collaborators

For numbered affiliations see end of article.

**Correspondence to**
Dr Toby O Smith;
toby.smith@uea.ac.uk

## ABSTRACT

**Objective** To test the effectiveness of a behaviour change physiotherapy intervention to increase physical activity compared with usual rehabilitation after total hip replacement (THR) or total knee replacement (TKR).

**Design** Multicentre, pragmatic, two-arm, open, randomised controlled, superiority trial.

**Setting** National Health Service providers in nine English hospitals.

**Participants** 224 individuals aged ≥18 years, undergoing a primary THR or TKR deemed 'moderately inactive' or 'inactive'.

**Intervention** Participants received either six, 30 min, weekly, group-based exercise sessions (usual care) or the same six weekly, group-based, exercise sessions each preceded by a 30 min cognitive behaviour discussion group aimed at challenging barriers to physical inactivity following surgery (experimental).

**Randomisation and blinding** Initial 75 participants were randomised 1:1 before changing the allocation ratio to 2:1 (experimental:usual care). Allocation was based on minimisation, stratifying on comorbidities, operation type and hospital. There was no blinding.

**Main outcome measures** Primary: University of California Los Angeles (UCLA) Activity Score at 12 months. Secondary: 6 and 12-month assessed function, pain, self-efficacy, kinesiophobia, psychological distress and quality of life.

**Results** Of the 1254 participants assessed for eligibility, 224 were included (139 experimental: 85 usual care). Mean age was 68.4 years (SD: 8.7), 63% were women, 52% underwent TKR. There was no between-group difference in UCLA score (mean difference: −0.03 (95% CI −0.52 to 0.45, p=0.89)). There were no differences observed in any of the secondary outcomes at 6 or 12 months. There were no important adverse events in either group. The COVID-19 pandemic contributed to the reduced intended sample size (target 260) and reduced intervention compliance.

**Conclusions** There is no evidence to suggest attending usual care physiotherapy sessions plus a group-based behaviour change intervention differs to attending usual care physiotherapy alone. As the trial could not reach its intended sample size, nor a proportion of participants receive their intended rehabilitation, this should be interpreted with caution.

**Trial registration number** ISRCTN29770908.

## STRENGTHS AND LIMITATIONS OF THIS STUDY

⇒ The multicentre recruitment approach enhanced external validity across population characteristics in England.
⇒ Functional, behavioural and psychological outcomes were collected to ensure a global participant assessment.
⇒ It was challenging to ensure that there were acceptable numbers of people in the group-based intervention.
⇒ All 12-month follow-up data were collected during the COVID-19 pandemic, potentially impacting on typical recovery and psychological outcomes.
⇒ The COVID-19 pandemic meant we were unable to reach our anticipated sample size or deliver the intervention as planned.

## INTRODUCTION

Total hip replacement (THR) and total knee replacement (TKR) are two highly successful orthopaedic procedures, which reduce pain for people with osteoarthritis.[1 2] Over 200 000 THRs and TKRs were performed in the United Kingdom (UK) in 2019 prepandemic.[1] Approximately 90% of patients are typically satisfied following THR and TKR,[2] with significant improvements in pain and physical function after 3–12 months.[2 3]

Historically, it has been assumed that people become more active following THR or TKR through the amelioration of joint pain.[4] However, current literature suggests physical

activity, at best, remains the same from preoperatively to post-operatively, and in some instances declines.[4 5]

People following THR and TKR have reported a number of challenges which make engaging in physical activity difficult, most notably psychosocial barriers and fear avoidance beliefs.[6] Such barriers include receiving insufficient and inconsistent information on being more physically active, fear of damaging joint replacements and causing pain and not being able to goalset or problemsolve physical activities within individual's lifestyles.[6] While previous international guidance has acknowledged the importance of physical activity on health and well-being, people following THR and TKR have reported difficulty in being active.[6] There is limited support or guidance currently offered on how to overcome these problems post-operatively.[6]

Not being physically active after joint replacement can have a major negative impact on a person's health and a burden on the National Health Service (NHS). Medical comorbidities are common in this population. These include hypertension (56%),[7] cardiovascular disease (20%),[8] diabetes (16%)[8] and multijoint pain (57%).[7] Approximately, 27% of people who undergo joint replacement have three or four comorbidities.[8] Medical comorbidities have a significant negative impact on both health-related quality of life (HRQoL) and result in a societal burden.[9 10] Participating in regular physical activity can decrease the risk of cardiovascular disease by 52%,[11] diabetes by 65%[12] and some cancers by 40%.[13] It is associated with a reduction in all-cause mortality by 33% and cardiovascular mortality by 35%.[14]

Current rehabilitation following THR and TKR in the UK, as advocated by the National Institute for Health and Care Excellence, centres around regaining joint movement, strength and gait re-education.[15] There is currently no evidence informing patients or healthcare professionals on how to increase physical activity specifically following THR and TKR. Following joint replacement, people have specific psychological needs and challenges, which differ to the non-joint replacement population.[6] Therefore, a specific intervention tailored to this population's health beliefs, including fear avoidance regarding implant survival, dislocation and increased knowledge on the impact of physical inactivity on other comorbidities, is required. Previous research has demonstrated that behaviour–change interventions can effectively increase physical activity across the lifespan.[16–20] Given this, it was hypothesised that such an intervention could be beneficial for this population. Accordingly, the purpose of this trial was to answer the research question 'following a primary THR or TKR, does a group exercise and behaviour-change intervention targeted to increase physical activity participation increase HRQoL and clinical outcomes over the initial 12 postoperative months compared with group exercise alone?'

## METHODS
### Study design
A full protocol has been published previously.[21]

This was a two-arm, open, pragmatic, parallel, multicentre, randomised controlled superiority trial. The study flowchart is presented as figure 1. Participants were recruited from eight UK NHS hospital trusts by the clinical team once they had been listed for THR or TKR. Interventions were delivered in physiotherapy departments within these NHS facilities.

We recruited adults who were due to undergo primary unilateral THR or TKR where the indication for surgery was degenerative joint pathology (not trauma). Potential participants were classified as 'moderately inactive' or 'inactive' using the General Practice Physical Activity Questionnaire[22] and have a Charlson Comorbidity Index (CCI) of ≥1 point.[23 24] We excluded people who were cognitively impaired, defined as an Abbreviated Mental Test Score (AMTS)[25] of <8; whose usual place of residence was a care home; were unable to read and/or comprehend English and had no access to a working telephone.

### Study treatments
Usual NHS surgical and in-patient care was received by both control and intervention groups. On hospital discharge, all participants attended 6 weekly, 30 min, group-based exercise classes within each hospital trust's physiotherapy department. These groups commenced within 4 weeks postoperation. The principles regarding prescription of group exercises to increase range of motion, strength and gait pattern, were consistent. While the rehabilitation of THR and TKR focuses on overall lower limb function, all participants following a THR focused on hip exercises, whereas those following a TKR focused on knee exercises. One physiotherapist (with or without a second physiotherapist or therapy assistant) ran each session.

The programme and rationale for the experimental intervention are presented in detail in online supplemental file 1. In brief, the intervention was grounded in the social cognitive theory[26] based on the theory that behaviour (physical activity level) is influenced by bidirectional relationships with personal factors (cognitive, emotional and physical) and environment. In this process, the cognitive behavioural approach in the PEP-TALK intervention used techniques to identify and target unhelpful thoughts and behaviours in order to produce adaptive thoughts, behaviours, emotions and physiological responses. Previous systematic reviews examining barriers and facilitators for older adults to increase physical activity have identified specific beliefs, which could reduce an individual's general self-efficacy.[4 6 27 28] These include: stigma, body image[28] and ageing stereotypes.[27] Unhelpful beliefs can be identified and explored using cognitive behavioural techniques to increase self-efficacy. The evidence also identified tools to increase general self-efficacy, which include the credibility of instructors and the information/physical activity tasks they provide.[27–29]

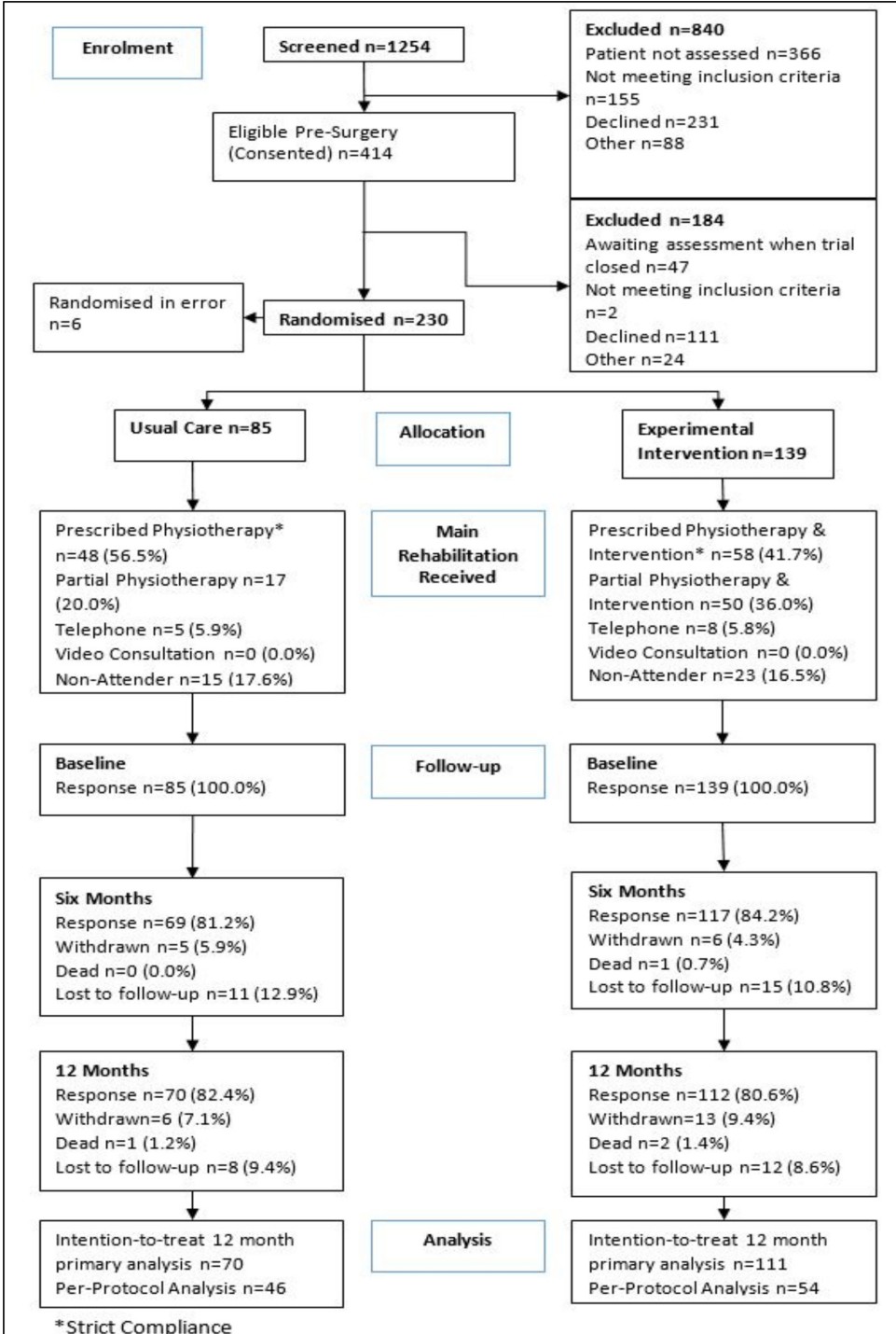

**Figure 1** CONSORT flowchart. CONSORT, Consolidated Standards of Reporting Trials.

The PEP-TALK intervention was designed to address these, exploring known barriers and facilitators to physical activity after joint replacement,[6] to promote increased participation in activity postoperatively.

In practice, participants randomised to the experimental group received the same 6 weekly, group-based, 30 min, exercise session as the usual care group. The only difference between the two groups was the addition of a 30 min, group-based, behaviour change intervention prior to the routine 30 min of exercise, and three

follow-up telephone calls 2, 4 and 6 weeks after the last group-based session. In the group-based sessions, participants were facilitated (as a group) to develop skills to overcome challenges to physical activity behaviour, supplemented through a workbook. This encouraged reflective activities such as recording physical, emotional and cognitive barriers and facilitators to physical activity. One physiotherapist (with or without a second physiotherapist or therapy assistant) ran each session. During the follow-up telephone calls, participant's goals were

reviewed, any barriers to the completion of these goals were identified, and the physiotherapist reviewed any 'unhelpful' and 'helpful' thoughts or feelings towards physical activity, which may have arisen since the last consultation and closed with the development of longer term physical activity goal setting. A treatment log was completed by physiotherapists to record the components of what was discussed across the group in each session and each telephone call.

Each member delivering the experimental intervention attended a 1-day training session, which taught the components and format of the intervention. To ensure compliance with the treatment protocol, the PEP-TALK team made regular visits for quality assurance.

### Data collection
At the time of enrolment, site healthcare professionals checked eligibility and recorded demographic characteristics. Baseline scores for outcome questionnaires were obtained before randomisation. Data collected at baseline included: gender, age, height and weight, CCI, self-reported presence and location of multisite joint pain, comorbidities determined from the medical notes, AMTS, employment status and occupation (when appropriate).

Participants were followed-up at 6 and 12 months after randomisation.

The primary outcome was the University of California Los Angeles (UCLA) Activity Score[30] (scored 0–10; higher scores indicate greater physical activity) at 12 months. This was selected as it is a reliable and valid self-reported tool to assess physical activity[31] and has been previously used for this means in orthopaedic trials.[32] Secondary outcomes at 6 and 12 months after randomisation were measured using the Lower Extremity Functional Scale (LEFS)[33] (scored 0 to 80, higher scores indicating less functional disability), Oxford Hip Score (OHS)[34] or Oxford Knee Score (OKS)[35] (scored 0–48, higher scores indicating less disease-specific function), Numerical Rating Scale (NRS) for pain (scored 0–10, higher scores indicating greater pain perception), the Generalized Self-Efficacy Scale (GSES)[36] (scored 10–40, higher scores indicating greater self-efficacy), the Tampa Scale for Kinesiophobia[37] (scored 17–68, higher scores indicating greater fear of motion), the Hospital Anxiety and Depression Scale (HADS)[38] (scored 0–21, higher scores indicating greater anxiety and depression), and the EQ-5D-5L[39] (scored 0–1, higher scores indicating greater HRQoL). Participants provided a retrospective assessment of complications at each 6-month follow-up period. Health resource utilisation data were collected but are not presented in this paper.

For each participant in the experimental intervention arm, the number of trial exercise sessions attended and group size of each session was recorded. The number of telephone contacts made after the end of the sessions and adherence with intervention protocols was also collected. There were no changes to the outcomes during the trial.

### Randomisation and masking
Random allocation was 1:1 originally. Randomisation was performed using a centralised computer randomisation programme provided by the Oxford Clinical Trials Research Unit (OCTRU). Research nurses and physiotherapists at recruiting centres enrolled participants and then assigned participants by accessing the online OCTRU randomisation programme, thereby adopting a concealed allocation approach. Randomisation was undertaken using a minimisation algorithm, stratified by: hospital site; type of joint replacement (THR or TKR); CCI of one to three versus ≥4.[23 24] It had a probabilistic element introduced to ensure unpredictability of treatment assignment.

The experimental intervention was designed to have three or more people per group.[21] Early sites found it difficult to consistently reach this level of participant numbers with the original 1:1 randomisation allocation. Accordingly, after 75 randomisations, we modified the randomisation ratio to 2:1 in favour of the experimental intervention. This ensured that a greater number of people are allocated to the experimental intervention. The sample size was increased to 260 to account for this change.

Masking participants or the teams providing interventions was not possible.

### Sample size
The trial was powered on the single primary outcome of UCLA at 12 months. Originally, 250 participants (125 per arm) were required to detect a standardised effect size of 0.4 with 80% power and 5% (two sided) significance, and allowing for 20% loss to follow-up. These calculations were based on the primary outcome, UCLA Activity Score at 12 months, assuming a baseline SD of 2.5 and a between-group difference of one.[32] The minimally clinically important difference (MCID) was reported as a within-person difference of 0.92 points.[32]

The target sample size was increased to 260 to account for the change in randomisation ratio.[21]

Results from the secondary outcomes provide supporting evidence for the results from the primary outcome analysis and are not powered separately. No allowance for multiple testing was included as a single primary outcome was considered.

### Statistical methods
There was no planned interim analyses or predefined stopping rules. Full analysis details are in the published statistical analysis plan.[40]

The primary outcome measure, UCLA at 12 months, was modelled using a linear mixed effect model adjusting for person within centre random effects, CCI, type or operation (TKR or THR), time (6 and 12 months) and baseline UCLA score as fixed effects using the intention-to-treat population (participants analysed as randomised). A treatment by time point interaction was included to allow time-specific treatment effects to be

calculated. This approach makes use of all available data at each time point. The secondary outcomes (LEFS, OKS, OHS, HADS, NRS for pain, GSES, Tampa, EQ-5D-5L Index and EQ-VAS) were analysed using a similar modelling approach. The number of participants with one or more complications were analysed using logistic regression, adjusting for minimisation factors and treatment. A total number of complications were analysed using Poisson Regression adjusting for the same factors.

Supporting analyses to the primary outcome included an area under the curve (AUC) analysis and complier average causal effect (CACE) analyses for all three predefined levels of compliance (Strict Compliance, Compliance, Attendance).[40] Full definitions of the three compliance levels are given in online supplemental file 2. The AUC analysis provided additional information on the trajectory of function recovery of these participants. The CACE analysis answered the question, for those participants who received the intervention as planned, did it improve function over usual care alone? The AUC analysis was performed using the same model as used for the primary analysis except including baseline UCLA Activity Score in the 'time'-fixed effect allowing time point-specific treatment effects to be calculated for baseline, 6 and 12 months. The CACE analysis has been performed through 10 000 bootstrapped samples. Adjusted linear regression was used for the 12-month UCLA Activity Score; adjusting for randomised treatment, baseline UCLA Activity Score, recruiting site, CCI (continuous) and joint replacement was used to obtain Intention To Treat (ITT) estimates. The pathway from treatment allocation to compliance (rate of potential compliers in the usual care group) was also estimated using adjusted linear regression: compliance indicators was analysed adjusting for the same variables. CACE estimates were obtained by taking the ratio of the ITT estimate and potential complier rate. SEs, CI and p values were calculated using the bootstrapped samples.

Other analyses examining the missing data assumptions, the per-protocol population, using a reduced model, treatment effects within predefined clinical subgroups and exploratory descriptive statistics for selected secondary outcomes by COVID-19.

## Study monitoring
A Trial Steering Committee (TSC) and Data Safety Monitoring Committee were appointed to independently review data on safety, protocol adherence and trial recruitment.

## Patient and public involvement
Patient involvement began during protocol development and continued throughout the trial. A patient-member (not enrolled in the trial) attended TSC meetings. The same patient-member was a coinvestigator. He provided insights into the trial conduct, particularly on data collection processes and helped interpret the findings to inform the trial's dissemination phase.

Trial participants who expressed an interest in receiving information on the trial findings were provided with this.

## RESULTS
### Recruitment and participant flow
Recruitment occurred between 12 April 2019 and 27 March 2020. The Consolidated Standards of Reporting Trial[41] flowchart is presented as figure 1. In total, 230 participants were randomised. Six were randomised in error, resulting in an analysable population of 224 participants (85 usual care; 139 experimental).

Due to the COVID-19 pandemic, 47 participants that had consented to take part in the study could not be randomised and the trial was stopped 30 participants short of its planned sample size. All elective THRs and TKRs were cancelled as part of the UK national COVID-19 lockdown (23 March 2020). Group-based physiotherapy classes within the participating hospital outpatient settings (a mechanism this trial relied on for both treatment groups) were also halted. Consequently, it was not feasible to continue the trial for the final 30 planned participants.

### Retention
The retention of participants is presented in figure 1. There were 37 withdrawals (13 usual care; 24 experimental). Online supplemental file 1 gives a summary of type of withdrawals by level of withdrawal and treatment group. The return of primary outcome data is presented in online supplemental table 2. This illustrates that for the primary, ITT, analysis of the UCLA Activity Score, there were 223 (99.6%) participants to supply a UCLA Activity Score at baseline (85 usual care; 138 experimental), 186 (83.0%) responses at 6 months (69 usual care; 117 experimental) and 181 (80.8%) responses at 12 months (70 usual care; 111 experimental).

### Participant characteristics
Baseline characteristics are presented by randomised treatment group in table 1. The mean participant age was 68.4 years (SD: 8.7), 62.9% were women with 52.2% undergoing TKR. Seventy-four per cent of the cohort had a CCI of 1–3 (mean 2.9 (SD: 1.3)). Mean BMI was 30.9 kg/m$^2$ (SD: 5.7). The mean duration of symptoms prior to surgery was 46.9 months (SD: 50.9) with 73.2% presenting with an American Society of Anesthesiology grade of 2 at surgery. As table 1 demonstrates, the two groups were comparable with the experimental group presenting with a slightly higher proportion of women (64.7% versus 60.0%), longer duration of symptoms (mean: 48.8 months versus 43.8 months) and fewer inactive participants (79.1% versus 83.5%) compared with the usual care group.

### Main analyses
The results of the analysis for the primary outcome measure are presented in table 2 and figure 2. There was

**Table 1** Baseline characteristics by randomised group

| | Usual (n=85) | Experimental (n=139) | Total (n=224) |
|---|---|---|---|
| Age, years | n=85, 68.5 (8.8) | n=139, 68.3 (8.6) | n=224, 68.4 (8.7) |
| UCLA Activity Score, 1–10 | n=85, 3.6 (1.5) | n=138, 3.6 (1.6) | n=223, 3.6 (1.5) |
| Joint replacement | | | |
| Hip replacement | 40 (47.1) | 67 (48.2) | 107 (47.8) |
| Knee replacement | 45 (52.9) | 72 (51.8) | 117 (52.2) |
| CCI, dichotomised | | | |
| 1–3 | 64 (75.3) | 102 (73.4) | 166 (74.1) |
| 4+ | 21 (24.7) | 37 (26.6) | 58 (25.9) |
| CCI, continuous | n=85, 2.8 (1.3) | n=139, 3.0 (1.3) | n=224, 2.9 (1.3) |
| Sex | | | |
| Female | 51 (60.0) | 90 (64.7) | 141 (62.9) |
| Male | 34 (40.0) | 49 (35.3) | 83 (37.1) |
| BMI, categories | | | |
| Healthy weight | 15 (17.6) | 25 (18.0) | 40 (17.9) |
| Overweight | 22 (25.9) | 45 (32.4) | 67 (29.9) |
| Obese | 42 (49.4) | 60 (43.2) | 102 (45.5) |
| Morbidly obese | 6 (7.1) | 9 (6.5) | 15 (6.7) |
| BMI, kg/m$^2$ | n=85, 31.1 (5.9) | n=139, 30.7 (5.6) | n=224, 30.9 (5.7) |
| Joint pain in the past 7 days | | | |
| Yes | 85 (100.0) | 138 (99.3) | 223 (99.6) |
| No | 0 (0.0) | 1 (0.7) | 1 (0.4) |
| GPPAQ level | | | |
| Active | 0 (0.0) | 0 (0.0) | 0 (0.0) |
| Moderately active | 2 (2.4) | 1 (0.7) | 3 (1.3) |
| Moderately inactive | 12 (14.1) | 28 (20.1) | 40 (17.9) |
| Inactive | 71 (83.5) | 110 (79.1) | 181 (80.8) |
| AMTS | n=85, 9.6 (0.6) | n=139, 9.6 (0.6) | n=224, 9.6 (0.6) |
| EQ-5D-5L score | n=85, 0.4 (0.2) | n=139, 0.4 (0.3) | n=224, 0.4 (0.2) |
| EQ-VAS, 0–100 | n=85, 61.3 (20.0) | n=139, 60.6 (23.6) | n=224, 60.9 (22.2) |
| Numeric pain, 0–10 | n=85, 6.9 (1.9) | n=139, 7.2 (1.8) | n=224, 7.1 (1.9) |
| Symptom duration, months | n=85, 43.8 (48.8) | n=138, 48.8 (52.2) | n=223, 46.9 (50.9) |
| ASA classification | | | |
| 1 | 4 (4.7) | 12 (8.6) | 16 (7.1) |
| 2 | 61 (71.8) | 103 (74.1) | 164 (73.2) |
| 3 | 20 (23.5) | 22 (15.8) | 42 (18.8) |
| 4 | 0 (0.0) | 2 (1.4) | 2 (0.9) |

Data are mean (SD) or n (%).+Stratification factor used in randomisation.
AMTS, Abbreviated Mental Test Score; ASA, American Society of Anesthesiologists; BMI, body mass index; CCI, Charlson Comorbidity Index; EQ-5D-5L, EuroQol 5-level ; EQ-VAS, EuroQol Visual Analogue Scale; GPPAQ, General Practice Physical Activity Questionnaire; UCLA, University of California, Los Angeles.

no evidence to support rejecting the null hypothesis that there was no difference between attending group-based exercise plus a group-based behaviour change intervention and attending group-based exercise alone on the UCLA Activity Score at 12 months postrandomisation, at the 5% significance level (mean difference: −0.03; 95% CI −0.52 to 0.45; p=0.89). However, as the trial could not reach its intended final sample size due to the COVID-19 pandemic, this result should be interpreted with caution. The interpretation of the results did not change on per-protocol analysis or reduced model analysis (online supplemental table 3; online supplemental table 4).

**Table 2** UCLA Activity Score (primary outcome) results

| Time point | Usual<br>n, mean (SD) | Experimental<br>n, mean (SD) | Mean difference<br>Unadjusted | Adjusted (95% CI) | P value |
|---|---|---|---|---|---|
| Baseline | n=85, 3.62 (1.52) | n=138, 3.57 (1.57) | –0.06 | – | – |
| 6 months | n=69, 4.77 (1.52) | n=117, 4.97 (1.68) | 0.20 | 0.27 (–0.21,0.76) | 0.27 |
| 12 months (primary outcome) | n=70, 4.87 (1.61) | n=111, 4.84 (1.91) | –0.03 | –0.03 (–0.52,0.45) | 0.89 |
| Area under the curve over 12 months | 4.81 (0.29) | 4.89 (0.28) | – | 0.09 (-0.47,0.64) | 0.88 |
| CACE: strict compliance | – | n=46 | – | –0.24 (–1.45,0.96) | 0.69 |
| CACE: compliance | – | n=58 | – | –0.20 (–1.19,0.79) | 0.69 |
| CACE: attendance | – | n=81 | – | –0.16 (–0.90,0.59) | 0.68 |

For the AUC analysis, the SD presented are the standard errors for these estimates calculated using the delta method. CACE analysis based on 10 000 bootstrapped samples.

AUC, area under the curve; CACE, complier average causal effect; n, number of participants; UCLA, University of California Los Angeles.

Three CACE estimations were performed on the 12 month UCLA Activity Score, one for each definition of compliance (Strict Compliance, Compliance and Attendance). Table 2 presents the CACE estimates for the three levels of compliance. There was no difference in outcome based on these analyses and all effect estimates were within the MCID of 0.92.[34]

The results of all continuous secondary outcomes are presented in table 3. They demonstrate no significant between-group differences for any of the continuous secondary outcomes at any time point. A general pattern of improvement from baseline to 6 months, then levelling off at 12 months with no significant between-group differences observable, was seen throughout.

A total of 141 complications were reported from 75 participants, 50 (35.5%) in the usual care group and 91 (64.5%) in the experimental group (table 4; online supplemental figure 1). It should be noted that 62.1% of participants were randomised to the experimental group, so this apparent difference is expected if complication rate was the same across both groups. The most common complications were increased pain either in the operated joint or in other joints, wound infections, medical complications and stiffness in the operated joint. Most complications (65.2%) were reported in the first 6 months of postrandomisation. There was no difference in the number of people who had a complication (28 versus 47; OR: 1.03; 95% CI 0.56 to 1.89) or total numbers of complications (50 versus 91; OR: 1.10; 95% CI 0.77 to 1.56) between the usual care and experimental group, respectively. There was one adverse event (fall, usual care) and three serious adverse events (two experimental (cardiac failure, pneumonia), one usual care (suspected deep vein thrombosis)).

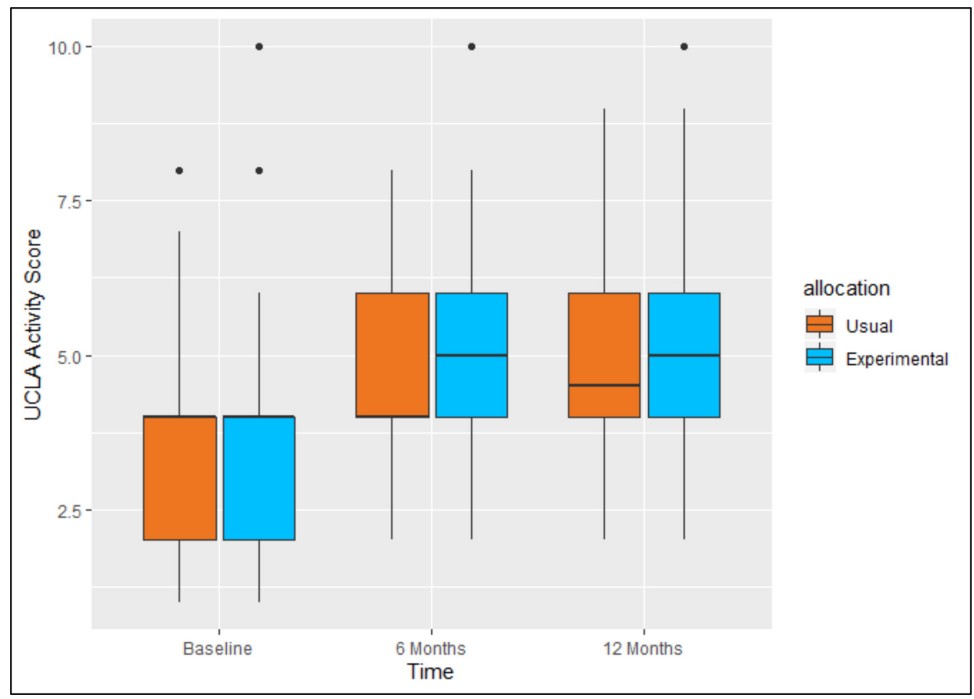

**Figure 2** UCLA Activity Score boxplots. UCLA, University of California Los Angeles.

**Table 3** Continuous secondary outcome results

| Time point | Usual<br>n, Mean (SD) | Experimental<br>n, Mean (SD) | Mean difference<br>Unadjusted | Adjusted (95% CI) | P value |
|---|---|---|---|---|---|
| **Lower Extremity Functional Scale** | | | | | |
| Baseline | n=82, 23.72 (13.11) | n=130, 24.50 (14.07) | 0.78 | – | – |
| 6 months | n=45, 45.40 (19.76) | n=80, 51.44 (17.70) | 6.04 | 2.60 (–3.29 to 8.50) | 0.39 |
| 12 months | n=51, 47.86 (18.97) | n=80, 50.67 (21.40) | 2.81 | 1.26 (–4.61 to 7.13) | 0.67 |
| **Oxford Hip Score** | | | | | |
| Baseline | n=40, 16.05 (6.36) | n=67, 16.78 (7.99) | 0.73 | – | – |
| 6 months | n=28, 34.84 (11.73) | n=50, 39.68 (8.93) | 4.84 | 3.86 (–0.92 to 8.64) | 0.11 |
| 12 months | n=27, 36.90 (12.48) | n=48, 39.42 (10.46) | 2.52 | 2.37 (–2.53 to 7.27) | 0.34 |
| **Oxford Knee Score** | | | | | |
| Baseline | n=45, 18.67 (8.51) | n=72, 17.46 (6.99) | −1.21 | – | – |
| 6 months | n=33, 35.20 (7.62) | n=51, 33.45 (9.38) | −1.75 | −1.74 (–5.03 to 1.54) | 0.30 |
| 12 months | n=35, 34.90 (8.46) | n=55, 33.54 (9.84) | −1.36 | −1.43 (–4.72 to 1.86) | 0.39 |
| **Numerical Rating Scale for Pain** | | | | | |
| Baseline | n=85, 6.87 (1.94) | n=139, 7.23 (1.79) | 0.36 | – | – |
| 6 months | n=61, 3.34 (2.59) | n=101, 3.54 (2.74) | 0.20 | 0.19 (–0.64 to 1.02) | 0.66 |
| 12 months | n=61, 4.08 (2.87) | n=102, 3.33 (2.85) | −0.75 | −0.75 (–1.59 to 0.09) | 0.08 |
| **Generalised Self-Efficacy Scale** | | | | | |
| Baseline | n=84, 31.31 (5.49) | n=138, 31.67 (5.39) | 0.36 | – | – |
| 6 months | n=58, 31.88 (5.18) | n=98, 33.03 (5.30) | 1.15 | 1.15 (–0.30 to 2.61) | 0.12 |
| 12 months | n=61, 32.16 (5.55) | n=101, 32.20 (6.72) | 0.03 | 0.33 (–1.13 to 1.78) | 0.66 |
| **Tampa Scale for Kinesiophobia** | | | | | |
| Baseline | n=85, 40.04 (7.44) | n=136, 39.77 (7.75) | −0.26 | – | – |
| 6 months | n=56, 35.77 (7.74) | n=91, 34.77 (7.29) | −1.00 | −0.39 (–2.40 to 1.61) | 0.70 |
| 12 months | n=57, 36.56 (6.91) | n=90, 35.06 (8.27) | −1.51 | −0.77 (–2.79 to 1.24) | 0.45 |
| **Hospital Anxiety and Depression Scale (overall)** | | | | | |
| Baseline | n=85, 11.85 (6.16) | n=138, 12.50 (7.07) | 0.65 | – | – |
| 6 months | n=59, 8.97 (6.52) | n=97, 8.81 (6.36) | −0.15 | −1.18 (–2.73 to 0.37) | 0.14 |
| 12 months | n=62, 9.02 (6.61) | n=98, 9.70 (6.99) | 0.69 | 0.52 (–1.03 to 2.06) | 0.51 |
| **Hospital Anxiety and Depression Scale (anxiety)** | | | | | |
| Baseline | n=85, 5.89 (3.78) | n=138, 6.63 (4.07) | 0.74 | – | – |
| 6 months | n=60, 4.95 (4.01) | n=98, 4.95 (3.57) | 0.00 | −0.71 (–1.67 to 0.25) | 0.15 |
| 12 months | n=62, 4.76 (3.73) | n=99, 5.46 (3.84) | 0.71 | 0.36 (–0.60 to 1.31) | 0.46 |
| **Hospital Anxiety and Depression Scale (depression)** | | | | | |
| Baseline | n=85, 5.95 (3.16) | n=139, 5.89 (3.81) | −0.06 | – | – |
| 6 months | n=61, 4.03 (3.27) | n=99, 3.90 (3.51) | −0.13 | −0.25 (–1.13 to 0.63) | 0.58 |
| 12 months | n=62, 4.26 (3.47) | n=101, 4.30 (4.02) | 0.04 | 0.24 (–0.65 to 1.12) | 0.60 |
| **EQ-5D-5L Index** | | | | | |
| Baseline | n=85, 0.40 (0.22) | n=139, 0.39 (0.27) | −0.01 | – | – |
| 6 months | n=68, 0.66 (0.23) | n=117, 0.69 (0.25) | 0.03 | 0.03 (–0.03 to 0.10) | 0.31 |
| 12 months | n=70, 0.67 (0.24) | n=113, 0.67 (0.29) | 0.00 | 0.00 (–0.06 to 0.07) | 0.93 |
| **EQ-VAS** | | | | | |
| Baseline | n=85, 61.33 (20.01) | n=139, 60.58 (23.56) | −0.75 | – | – |
| 6 months | n=68, 70.93 (18.67) | n=117, 73.86 (20.02) | 2.94 | 2.84 (–2.31 to 7.99) | 0.28 |
| 12 months | n=69, 72.51 (17.90) | n=110, 72.94 (19.98) | 0.43 | 1.47 (–3.73 to 6.68) | 0.58 |

EQ-5D-5L, EuroQol 5-level; EQ-VAS, EuroQol Visual Analogue Scale.

**Table 4** Complication results

| | Usual | Experimental | OR | |
| | N (%) | N (%) | (95% CI) | P value |
|---|---|---|---|---|
| Number of participants who had a complication | 28 (32.94) | 47 (33.81) | 1.03 (0.56 to 1.89) | 0.94 |
| Total complications | 50 (58.82) | 91 (65.47) | 1.10 (0.77 to 1.56) | 0.61 |

### Analysis by compliance

Treatment compliance is summarised in online supplemental figure 2. Compliance is reported by categories as defined in the analysis plan.[40] In total, 489 experimental intervention or physiotherapy exercise sessions were held. The sessions ran from 8 May 2019 to 18 March 2020. Of 162 were experimental sessions and 327 were exercise alone sessions (161 usual care; 166 experimental). There was one experimental class that was not accompanied by a physiotherapy class.

A major component of the definition of compliance for the experimental group was the group class sizes. The median class size for the intervention classes was two with a range of 1–14. Online supplemental figure 3 is a plot of the group sizes for all intervention sessions. Any class with three or more participants was considered a 'compliant' class. In total, 75 (46.3%) of the 162 intervention sessions had three or more participants. To address the issue of compliance, the randomisation procedure was changed from 1:1 to 2:1. Online supplemental figure 4 is a breakdown of treatment compliance by participants randomised using either a 1:1 or 2:1 randomisation ratio. In both groups, the number of participants who were non-compliant rose considerably and the number of strict compliers fell after the change from 1:1 to 2:1 randomisation. A confounder to this result is that participants whose intervention was disrupted by COVID-19 were all randomised using a 2:1 ratio. The large increase in non-compliance in that population is seen in online supplemental figure 4.

### Impact of COVID-19 on trial findings

The level of disruption to the intervention delivery caused by the COVID-19 pandemic was high. There was a high level of non-compliance, particularly in the experimental group. This apparent between-group difference in non-compliance was because the predefined definitions of compliance were stricter in the experimental than the usual care group. To be an 'Attender' in the experimental group, one needed to attend four out of six group intervention sessions, to achieve the same level of compliance in the usual care group, only one session was required to be attended. In the usual care group, 66 (77.6%) attended at least one physiotherapy session, a similar proportion, 111 (80%), attended at least one physiotherapy session in the experimental group. Due to the added therapy the experimental group received, the definition for compliance had to be stricter, but both groups had a similar proportion who attended at least one session.

The final months of the trial, before all group-based physiotherapy classes within the hospital outpatient setting were halted due to the COVID-19 pandemic, yielded the highest group sizes. online Supplementary Figure 4 summarises the compliance to the experimental group by pre-COVID-19 compared with COVID-19 to estimate the impact of the pandemic on compliance. This is plotted by time in online supplementary file 3. Based on this, a large proportion of participants who could not be randomised due to the trial closure would have ended up falling into either the 'Compliant' or 'Strict Compliant' groups.

### Additional analyses

The missing data analysis suggests that the missing at random assumption made in the primary analysis is appropriate (online supplementary figure 5). The per-protocol and reduced model results support the main findings from the trial and there was no evidence of any difference in the exploratory subgroup analysis. The exploratory descriptive statistics by COVID-19 status may suggest participants in the COVID-19 group had poorer psychological outcomes (online supplementary table 5). The results are presented in full in online supplemental figure 6.

### DISCUSSION

The findings suggest that following THR or TKR, there is no difference between the addition of a group-based exercise and behaviour change intervention in physical activity and other clinical outcomes during the first postoperative year compared with attending group-based exercise alone. However, the COVID-19 pandemic significantly impacted on this trial, whereby the intended sample size was not achieved, and a considerable proportion of participants were unable to receive their intended postoperative rehabilitation. Accordingly, these findings should be interpreted with caution.

The rationale for undertaking this study was the uncertainty over how to increase physical activity following THR and TKR. While several studies have been published over the intervening period acknowledging that physical activity remains low following joint replacement,[42–44] there continues to be uncertainty over how to overcome this. Studies in other populations, most notably older adults, individuals with chronic respiratory disorders and those with chronic rheumatological diseases have provided promise that a behaviour change intervention

may improve physical activity.[17–20] As previously acknowledged, the specific challenges which individuals face in relation to fear avoidance, beliefs about implant failure, multijoint pain and other comorbidities[6] may account for why this behaviour change intervention did not demonstrate similar changes. However, this trial specifically relates to the effectiveness of a behaviour intervention targeted to the behaviour change construct of self-efficacy in the joint replacement population. There may remain value for future research exploring the effectiveness of other behaviour change constructs, to increase physical activity after these orthopaedic procedures. Furthermore, the results from this trial have been impacted by the COVID-19 pandemic, principally on intervention delivery and compliance. Given the impact COVID-19 had, there still remains a need to better understand how to increase physical activity following THR or TKR.

Trial participants understood the research objective was to explore the effectiveness of an intervention aimed at increasing physical activity following THR or TKR. However, compliance to the intervention was low throughout the trial. Accordingly, the appetite to increase physical activity remains uncertain. Previous literature has suggested that while individuals may be no more physically active after joint replacement,[44 45] clinical outcomes and specifically pain do significantly improve.[46 47] This corresponds with an improvement in HRQoL. Patient satisfaction to outcome and expectations may be met, but this is not translated into increased physical activity. Given the wider health benefits which physical activity confers, consideration should be made on how health professionals promote physical activity messages within postoperative recovery programmes, so added health gains are maximised. How this is operationalised following this trial's findings remains unclear.

While the results indicate no superiority to the addition of a behaviour change intervention to usual physiotherapy rehabilitation after TKR or THR, the findings offer important clinical implications. First, the trial indicates that joint replacement and usual physiotherapy rehabilitation can improve clinical outcomes. Previous literature suggests improvements in pain, function and HRQoL[46 47] for people following THR and TKR. However, the trial also indicates both pre-COVID-19 and post-COVID-19 that there were differences in adherence and compliance to both usual and experimental physiotherapy interventions. While previous literature has highlighted geographical and service-provision differences in rehabilitation after joint replacement,[48 49] there has been limited evidence to indicate variability in adherence to rehabilitation. This may reflect variation in rehabilitation need. While some patients may need substantial levels of physiotherapy following joint replacement to promote physical function, activity and improvements in HRQoL, these may not be homogeneous within the population.[50] Stratification on rehabilitation need may, therefore, be warranted. While previous authors have attempted to identify those at most risk of poor outcomes postoperative,[51 52] there remains uncertainty over what physiotherapy intervention is more beneficial for these patients. Further consideration on the optimal rehabilitation programme to promote physical activity for those with the most to gain as opposed to assuming all, as adopted in this trial, may be indicated.

There are several trial strengths and limitations to be considered. A major strength was the pragmatic approach taken to assess effectiveness. The broad eligibility criteria to reflect typical patients who undergo THR and TKR, balanced by the inclusion of only those, who were preoperatively moderately inactive or inactive, meant the eligibility criteria were constructed to theoretically recruit those who had the most to gain. The multisite, national recruitment process across NHS health trusts also offered the ability to recruit a diverse cohort in relation to socioeconomic, ethnic and geographical factors. However, a limitation to the design was that several measures which may have characterised such diversity including level of deprivation, educational status, ethnicity and educational background were not collected. This decision was made to offer a more efficient data collection process, not overburdening participants with extensive demographic data requests. Smith *et al*[53] previously acknowledged that this as a recurrent limitation to musculoskeletal research. Future research should consider the impact of socioeconomic and deprivation factors both on the design of interventions, processes and analysis. A further limitation was the impact of COVID-19. While acknowledged that the trial over-recruited, consenting 277 participants, only 230 were randomised as the pandemic disrupted surgical and rehabilitation delivery. This means that the results were underpowered to answer the trial's primary research question. Second, 69 individuals who were receiving rehabilitation during this time had their intervention delivery impacted on this change in service provision. Consequently, intervention compliance reduced, impacting on any effect estimate generated from that point onwards. Given this equated to 30% of the cohort, it is proposed this had a significant impact. What is more difficult to estimate is the impact of the COVID-19 social restrictions on outcome. All participants experienced the 2020 social restrictions prior to completing their 12-month questionnaires (first 12-month questionnaire completed 23 March 2020). While previous studies[54 55] indicate that individuals with joint pain substantially reduced their natural physical activity engagement during this time, we did not specifically collect data to ascertain the effects of 'lockdown' on outcomes. The effect of this on 12-month results should, therefore, be considered.

## CONCLUSIONS

The addition of a group-based behaviour change intervention to usual physiotherapy rehabilitation following primary THR and TKR does not offer benefit over usual physiotherapy alone on physical activity and clinical outcomes over the first 12 postoperative months. These

findings should be viewed with caution as the COVID-19 pandemic impacted on both the ability of participants to undergo joint replacement and compliance to their rehabilitation. Given the health and social benefits which being active offer older adults, further exploration on methods to increase physical activity for those who are inactive following joint replacement remains important.

**Author affiliations**
[1] Nuffield Department of Orthopaedics, Rheumatology and Musculoskeletal Sciences, University of Oxford, Oxford, UK
[2] School of Health Sciences, University of East Anglia, Norwich, UK
[3] Centre for Statistics in Medicine, University of Oxford, Oxford, UK
[4] Trauma and Orthopaedic Department, St George's University Hospitals NHS Foundation Trust, London, UK
[5] College of Medicine and Health, University of Exeter, Exeter, UK

**Collaborators** PEP-TALK Collaborators: Mr Steve Algar (Banbury, PPI Representative), Dr Zara Hansen (ZH) (University of Oxford), Dr Vicki Barber (Oxford Clinical Trials Research Unit (OCTRU), University of Oxford), Mr Malcolm Hart (OCTRU), University of Oxford), May Ee Png (OCTRU, University of Oxford), Professor Karen Barker (Principal Investigator—Oxford University Hospital NHS Foundation Trust (OUH), Mr Ian Smith (Principal Investigator—Lewisham and Greenwich University Hospitals NHS Foundation Trust), Professor Iain McNamara (Principal Investigator (Norfolk and Norwich University Hospitals NHS Foundation Trust (NNUH) & Spire Norwich), Mr Michael Dunn (Principal Investigator - St George's University Hospital NHS Foundation Trust), Mrs Dawn Lockey (Principal Investigator— City Hospitals Sunderland NHS Foundation Trust), Mr Sonny Driver (Principal Investigator—North Middlesex University Hospital NHS Trust), Mrs Jamila Kassam (Barts Health NHS Trust), Mr Peter Penny (Norwich Spire, Norwich), Mrs Celia Woodhouse, Mrs Tracey Potter and Mrs Helena Daniell (NNUH), Mr Alex Herring and Mrs Yan Cunningham (City Hospitals Sunderland NHS Foundation Trust), Irrum Afzal (South West London Elective Orthopaedic Centre), Mr Maninderpal Matharu (Barts Health NHS Trust) and Mrs Tamsin Hughes, Ms Erin Hannink and Mrs Michelle Moynihan (OUH).

**Contributors** TOS, BF, SD, SL, AO, CH, AG, SP researched the topic and devised the study. TOS, SP, AO, BF, SD, CH, AG, SL provided the first draft of the manuscript. AO, SD provided statistical analysis and oversight. TOS, SP, BF, AO, SD, CH, AG, SL contributed equally to manuscript preparation. TS acts a guarantor.

**Funding** The research is supported by the National Institute for Health Research (NIHR) Research for Patient Benefit grant (PB-PG-1216-20008). The views expressed are those of the author(s) and not necessarily those of the NHS, the NIHR or the Department of Health and Social Care.

**Competing interests** None declared.

**Patient and public involvement** Patients and/or the public were involved in the design, or conduct, or reporting, or dissemination plans of this research. Refer to the Methods section for further details.

**Patient consent for publication** Not applicable.

**Ethics approval** This study involves human participants and was approved by NHS NRES South Central Oxford B (18/SC/0423). Participants gave informed consent to participate in the study before taking part.

**Provenance and peer review** Not commissioned; externally peer reviewed.

**Data availability statement** Data are available upon reasonable request. Access to the de-identified dataset for purposes of research other than this study, would be at the discretion of the Chief Investigator, Dr Toby Smith and OCTRU. Requests for the de-identified dataset generated during the current study should be made to the Chief Investigator, Dr Toby Smith (email: toby.smith@uea.ac.uk) or OCTRU ( octrutrialshub@ndorms.ox.ac.uk). Dr Toby Smith and OCTRU will consider requests once the main results from the study have been published up until 31 December 2026.

of the author(s) and are not endorsed by BMJ. BMJ disclaims all liability and responsibility arising from any reliance placed on the content. Where the content includes any translated material, BMJ does not warrant the accuracy and reliability of the translations (including but not limited to local regulations, clinical guidelines, terminology, drug names and drug dosages), and is not responsible for any error and/or omissions arising from translation and adaptation or otherwise.

**ORCID iD**
Beth Fordham http://orcid.org/0000-0001-5996-3563

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
