## [Reviewer comments · BMJ Open]

ARTICLE DETAILS

TITLE (PROVISIONAL)	Randomised controlled trial of a behaviour change physiotherapy intervention to increase physical activity following hip and knee replacement: the PEP-TALK trial
AUTHORS	Smith, Toby O.; Parsons, Scott; Ooms, Alexander; Dutton, Susan; Fordham, Beth; Garrett, Angela; Hing, Caroline; Lamb, Sarah

VERSION 1 – REVIEW

REVIEWER	Hawke, Lyndon J. La Trobe University
REVIEW RETURNED	08-Mar-2022

GENERAL COMMENTS	Behaviour change physiotherapy intervention to increase physical activity following hip and knee replacement: the PEP-TALK randomised controlled trial Comments to the author General comments This was a multicentre, pragmatic, two-arm, open randomised controlled superiority trial evaluating the effectiveness of a behavioural intervention to increase physical activity after joint replacement. This trial was thoroughly planned, well-conducted and rigorous trial - it is such a shame that the Covid pandemic interrupted your hard work. This is an important yet under-investigated area of orthopaedic rehabilitation. Despite the limitation of the trial being underpowered relative to the sample size estimation, the trial outcomes still provide valuable estimates of effect and insights into the impact of behavioural interventions on physical activity after joint replacement, and which behavioural interventions are, or are not, effective. The manuscript is well written and tells an interesting story. The figures and tables are comprehensive, well set out and complement the manuscript. The supplementary information also provides comprehensive and valuable background information. Specific comments Title
---

	Adequately descriptive. No changes suggested. Abstract Comprehensively summarises key findings and conclusions. No changes suggested. Introduction Research questions clearly stated. Gaps in understanding highlighted and how this study hopes to address the gaps is explained. Good logical flow of ideas culminating in research questions. All major introductory discussion points of this topic e.g. assumptions reduced pain after joint replacement would lead to increased activity, health impacts of physical inactivity in this population, behaviour intervention thought to be the best way forward etc covered. No changes suggested Methods Study design Suitable overall design A strength is how this study targeted people who were inactive (using the GPPAQ) – I don't believe this has been done before in this population. Study treatments A strength is how this study was conducted with people early in their recovery (4 weeks after surgery) within their 'teachable moment' phase. While the rationale for the intervention is fully described in Supplementary File 1, I believe the manuscript would benefit from a brief mention of the intervention rationale, especially highlighting that the intervention was grounded in theory, informed by a methodological framework and targeted the behavioural change construct of self-efficacy. This adds weight to your intervention, particularly if readers of this article have not read the published protocol. Page 6, line 15. Suggest replacing the word 'we' with 'the PEP-TALK team'. Data collection The manuscript could be strengthened with a brief rationale for your choice of primary outcome (UCLA Activity Score) to measure the construct of physical activity e.g. why did you use a self-reported outcome measure instead of a more objective measure like accelerometry, or a more comprehensive self-report measure such as IPAQ, and comment briefly as to whether the UCLA
--	--

	Activity Score has been validated to measure physical activity in this population - perhaps including the following reference (or similar reference) may be of value. Naal, Florian D., Franco M. Impellizzeri, and Michael Leunig. "Which is the best activity rating scale for patients undergoing total joint arthroplasty?." Clinical orthopaedics and related research 467.4 (2009): 958-965. Randomisation and masking Well described. No changes suggested. Sample size Well described and appropriate for this study. No changes suggested. Statistical methods No changes suggested Study monitoring No changes suggested Patient and public involvement No changes suggested Results Comprehensively reported. Good flow between text and figures/tables Table 1: ASA Classification. I think the 'ASA' abbreviation definition is missing; e.g. ASA=American Society of Anesthesiologists Discussion The study findings are clearly summarised as is the gap the study intends to fill. Paragraph 2, page 13 line 31 'However, as previously acknowledged, the specific challenges which individuals face in relation to fear avoidance, beliefs about implant failure, multi-joint pain and other comorbidities[6] may account for why this behaviour change intervention did not demonstrate similar changes.'
--	--

	Suggest adding a sentence commenting on what your study results say about the effectiveness of behaviour interventions that target the behaviour change construct of self-efficacy in the joint replacement population, and the value of future research exploring the effectiveness of other behaviour change constructs to increase physical activity after joint replacement. Strengths and limitations Well articulated and contextually explained. Conclusions Concise and on-point. No changes suggested.
--	--

REVIEWER	Cameron, Claire University of Otago, Dunedin School of Medicine
REVIEW RETURNED	08-Mar-2022

GENERAL COMMENTS	Thank you for the opportunity to review this manuscript. The purpose of the study is to test the effectiveness of a behaviour change physiotherapy intervention to increase physical activity compared with usual rehabilitation after Total Hip Replacement (THR) or Total Knee Replacement (TKR). It is well prepared and thorough. However, I think clarification is needed to make the manuscript more readable.. I appreciate that there is a published protocol and analysis plan out there, but I think that this paper ought to stand alone in the sense that the reader should be able to understand what is being described and seek further detail in those other documents if need be. It is commendable that the trial was completed and written up during the COVID-19 pandemic. I know that that has created many, sometimes, insurmountable obstacles.  1. The abstract is a very good summary of the study. I think the "DESIGN" section needs more commas. I also think the 'INTERVENTION' section should indicate what the duration of the intervention was (12 months). 2. This is described as a superiority trial. What makes it a superiority trial? Is this just signalling that it is neither an equivalency trial nor a non-inferiority trial? 3. Has this been registered as a trial? Somewhere that information should be included. 4. 'Randomisation and Masking', page 6: Was there concealment of allocation at the time of enrolment? How did that process work? What were the minimisation factors (minimisation being different to stratification)? Was the change in randomisation ratio a change that was decided on a priori? What would be the implications of this change? 5. 'Sample Size', page 7. This calculation is a simple power calculation which I assume is based on a simple comparison between the two groups for the primary outcome (UCLA activity score) at 12 months. I also assume this is a conservative estimate
--

	and may cover the estimation through a regression model. However, is it sufficiently powered for looking at interactions or the secondary outcomes you have listed? 6. The 'Statistical methods' section, page 7, could be clearer. For example, "Linear mixed effects models were used for the primary and continuous secondary outcomes". Presumably, the primary outcome is continuous. Could you list the secondary outcomes you are referring to? I have looked at the published analysis plan and it is very clearly laid out. I think that this section would benefit from some thought as to how to reduce the information from the plan into this short section in a way that it is clear. 7. What is the meaning or importance of the AUC analysis or the CACE analysis? How does this contribute to answering your research question? 8. 'Patient and Public Involvement' So, this is a patient who is not enrolled in the trial? There is no mention of a member of the public in this paragraph. This need some clarification. 9. There are many, many results in this paper. Why are there estimates and confidence intervals for the baseline and 6 month results when we are, primarily interested in the 12 month activity level post study? There are 11 regression analyses reported for the secondary outcomes - all at each time point. Is this a useful addition to the manuscript? Who do you want to read this? What message do you want them to receive? I would like some clarification on why there are so many results reported. Alongside all these results are a plethora of p-values which seems unnecessary if you have effect sizes and their precision (see CONSORT, point 17a and 17b). 10. The Discussion (page 13) is well written and concise. It clearly answers the research question and provides insight into the salient points of the results. 11. I recommend that you present relative frequencies in Supplementary Figure 1. On the face of it, it looks like there will be bias introduced by the over representation in the experimental group of various complications. However, the imbalance is due to the imbalance of the groups.
--	--

VERSION 1 – AUTHOR RESPONSE

Reviewer 1

Comment: General comments: This was a multicentre, pragmatic, two-arm, open randomised controlled superiority trial evaluating the effectiveness of a behavioural intervention to increase physical activity after joint replacement. This trial was thoroughly planned, well-conducted and rigorous trial - it is such a shame that the Covid pandemic interrupted your hard work. This is an important yet under-investigated area of orthopaedic rehabilitation. Despite the limitation of the trial being underpowered relative to the sample size estimation, the trial outcomes still provide valuable estimates of effect and insights into the impact of behavioural interventions on physical activity after joint replacement, and which behavioural interventions are, or are not, effective.

Response: Thank you. No amendment required.

Comment: The manuscript is well written and tells an interesting story. The figures and tables are comprehensive, well set out and complement the manuscript. The supplementary information also provides comprehensive and valuable background information.

Response: Thank you. No amendment required.

Specific comments

Title

- Adequately descriptive.

Response: Thank you. This has been updated in-line with the Editor's request above.

- No changes suggested.

Response: Thank you. No amendment required.

Abstract

- Comprehensively summarises key findings and conclusions.

Response: Thank you. No amendment required.

- No changes suggested.

Response: Thank you. No amendment required.

Introduction

- Research questions clearly stated.

Response: Thank you. No amendment required.

- Gaps in understanding highlighted and how this study hopes to address the gaps is explained.

Response: Thank you. No amendment required.

- Good logical flow of ideas culminating in research questions. All major introductory discussion points of this topic e.g. assumptions reduced pain after joint replacement would lead to increased activity, health impacts of physical inactivity in this population, behaviour intervention thought to be the best way forward etc covered.

Response: Thank you. No amendment required.

- No changes suggested

Response: Thank you. No amendment required.

Methods - Study design

- Suitable overall design

Response: Thank you. No amendment required.

- A strength is how this study targeted people who were inactive (using the GPPAQ) – I don't believe this has been done before in this population.

Response: Thank you. No amendment required.

Study treatments

- A strength is how this study was conducted with people early in their recovery (4 weeks after surgery) within their 'teachable moment' phase.

Response: Thank you. No amendment required.

- While the rationale for the intervention is fully described in Supplementary File 1, I believe the manuscript would benefit from a brief mention of the intervention rationale, especially highlighting that the intervention was grounded in theory, informed by a methodological framework and targeted the behavioural change construct of self-efficacy. This adds weight to your intervention, particularly if readers of this article have not read the published protocol.

Response: As requested, we have provided further information on the intervention rationale, based on the recommended signposting on underlying theory, promotion of self-efficacy (Methods, Study treatments, Paragraph 2, Lines 2-17).

- Page 6, line 15. Suggest replacing the word 'we' with 'the PEP-TALK team'.

Response: Revised as suggested (Methods, Study treatments, Paragraph 4, Line 3).

Data collection

- The manuscript could be strengthened with a brief rationale for your choice of primary outcome (UCLA Activity Score) to measure the construct of physical activity e.g. why did you use a self-reported outcome measure instead of a more objective measure like accelerometry, or a more comprehensive self-report measure such as IPAQ, and comment briefly as to whether the UCLA Activity Score has been validated to measure physical activity in this population - perhaps including the following reference (or similar reference) may be of value. Naal, Florian D., Franco M. Impellizzeri, and Michael Leunig. "Which is the best activity rating scale for patients undergoing total joint arthroplasty?." *Clinical orthopaedics and related research* 467.4 (2009): 958-965.

Response: We have provided further reasoning for the use of the UCLA as recommended (Methods, Data collection, Paragraph 3, Lines 2-4).

Randomisation and masking

- Well described.

Response: Thank you. No amendment required.

- No changes suggested.

Response: Thank you. No amendment required.

Sample size

- Well described and appropriate for this study.

Response: Thank you. No amendment required.

- No changes suggested.

Response: Thank you. No amendment required.

Statistical methods

- No changes suggested

Response: Thank you. No amendment required.

Study monitoring

- No changes suggested Patient and public involvement

Response: Thank you. No amendment required.

- No changes suggested

Response: Thank you. No amendment required.

Results

- Comprehensively reported.

Response: Thank you. No amendment required.

- Good flow between text and figures/tables

Response: Thank you. No amendment required.

- Table 1: ASA Classification. I think the 'ASA' abbreviation definition is missing; e.g. ASA=American Society of Anesthesiologists

Response: Thank you. This has now been added (Table 1, Footnote, ASA).

Discussion

- The study findings are clearly summarised as is the gap the study intends to fill.

Response: Thank you. No amendment required.

- Paragraph 2, page 13 line 31 'However, as previously acknowledged, the specific challenges which individuals face in relation to fear avoidance, beliefs about implant failure, multi-joint pain and other comorbidities[6] may account for why this behaviour change intervention did not demonstrate similar changes.' Suggest adding a sentence commenting on what your study results say about the effectiveness of behaviour interventions that target the behaviour change construct of self-efficacy in the joint replacement population, and the value of future research exploring the effectiveness of other behaviour change constructs to increase physical activity after joint replacement.

Response: This has been added, as suggested (Discussion, Paragraph 2, Lines 9-13).

Strengths and limitations

- Well articulated and contextually explained.

Response: Thank you. No amendment required.

Conclusions

- Concise and on-point.

Response: Thank you. No amendment required.

- No changes suggested

Response: Thank you. No amendment required.

Reviewer 2

Comment: Thank you for the opportunity to review this manuscript. The purpose of the study is to test the effectiveness of a behaviour change physiotherapy intervention to increase physical activity compared with usual rehabilitation after Total Hip Replacement (THR) or Total Knee Replacement (TKR). It is well prepared and thorough. However, I think clarification is needed to make the manuscript more readable. I appreciate that there is a published protocol and analysis plan out there, but I think that this paper ought to stand alone in the sense that the reader should be able to understand what is being described and seek further detail in those other documents if need be. It is commendable that the trial was completed and written up during the COVID-19 pandemic. I know that that has created many, sometimes, insurmountable obstacles.

Response: Thank you for your supportive words. We have followed the recommendations made by yourself and Reviewer 1 and provided further information on the trial, whilst respecting the journal's recommended word limits for this style of publication. Please find the amendments and revisions listed below.

Comment: 1. The abstract is a very good summary of the study. I think the "DESIGN" section needs more commas. I also think the 'INTERVENTION' section should indicate what the duration of the intervention was (12 months).

Response: We have added in further commas to describe the design (Abstract, Design, Line 1). The intervention was over 6 weeks. This has been described as 'six, 30-minute, weekly, group-based exercise sessions'. We have not revised the text as feel it is sufficiently detailed to ensure the duration of the intervention is documented (Abstract, Intervention, Lines 1-2).

Comment: 2. This is described as a superiority trial. What makes it a superiority trial? Is this just signalling that it is neither an equivalency trial nor a non-inferiority trial?

Response: The trial was designed to be powered in-order to assess whether the PEP-TALK trial intervention was superior to usual care, rather than being powered to assess equivalence. This was justified as the intervention, given that it is offering a treatment which is above usual care, would need to demonstrate superior results to be considered commissionable for service delivery in this setting. Through this reasoning, it would not have been appropriate to design this as an equivalency trial. In this instance, we feel the detailing of the design, rather than specific justification for this, is appropriate and have therefore elected not to dedicate word count on explaining this to the reader. The CONSORT reporting guidance encourages the reporting of design, hence we have adopted this approach. However, if the reviewer and editorial team feel strongly against this, we would happily reconsider.

Comment: 3. Has this been registered as a trial? Somewhere that information should be included.

Response: This trial was registered with ISRCTN prior to commencing. The trial registration number is presented in the Abstract (Abstract, Trial Registration: ISRCTN29770908).

Comment: 4. 'Randomisation and Masking', page 6: Was there concealment of allocation at the time of enrolment? How did that process work? What were the minimisation factors (minimisation being different to stratification)? Was the change in randomisation ratio a change that was decided on a priori? What would be the implications of this change?

Response: We have provided further details that this was a concealed allocation approach (Methods, Randomisation and masking, Paragraph 1, Line 4). The minimisation factors of hospital site, type of replacement and Charleston Comorbidity Index are stated (Methods, Randomisation and masking, Paragraph 1, Line 5-6). As suggested by the reviewer, further details on the change in randomisation ratio has now been provided (Methods, Randomisation and masking, Paragraph 2, Lines 1-6).

Comment: 5. 'Sample Size', page 7. This calculation is a simple power calculation which I assume is based on a simple comparison between the two groups for the primary outcome (UCLA activity score) at 12 months. I also assume this is a conservative estimate and may cover the estimation through a regression model. However, is it sufficiently powered for looking at interactions or the secondary outcomes you have listed?

Response: Further clarification on the sample size calculation and how it relates to the secondary outcomes has been incorporated into the revisions (Methods, Sample size, Paragraph 1, Line 1; Methods, Sample size, Paragraph 3, Lines 1-3).

Comment: 6. The 'Statistical methods' section, page 7, could be clearer. For example, "Linear mixed effects models were used for the primary and continuous secondary outcomes". Presumably, the primary outcome is continuous. Could you list the secondary outcomes you are referring to? I have looked at the published analysis plan and it is very clearly laid out. I think that this section would benefit from some thought as to how to reduce the information from the plan into this short section in a way that it is clear.

Response: We have provided further clarification and detail on the analysis based on this recommendation (Methods, Statistical methods, Paragraph 2, Lines 1-9).

Comment: 7. What is the meaning or importance of the AUC analysis or the CACE analysis? How does this contribute to answering your research question?

Response: AUC analysis looks at the overall response of the participants over the whole 12 month period, rather than examining a single time point. It provides additional information on the trajectory of function recovery of these participants. The CACE analysis provides an answer to the question, for those participants who received the intervention as planned, did it improve function over usual care alone? Compliance with both the intervention and usual care were impacted by the COVID-19 pandemic. Using different definitions of compliance gives us the most information about any actual impact of the intervention on the outcome measure. This also enabled us to explore the impact of COVID-19 on these patients. We have provided this detail in the Statistical Methods (Methods, Statistical analysis, Paragraph 3, Line 3-5).

Comment: 8. 'Patient and Public Involvement' So, this is a patient who is not enrolled in the trial? There is no mention of a member of the public in this paragraph. This need some clarification.

Response: We have provided greater clarity on this, emphasising that our Patient and Public Involvement team member was not enrolled in the trial (Methods, Patient and Public Involvement, Paragraph 1, Line 2) and separating out his involvement to the trial participants receiving the findings, section (Methods, Patient and Public Involvement, Paragraph 2, Lines 1-2).

Comment: 9. There are many, many results in this paper. Why are there estimates and confidence intervals for the baseline and 6-month results when we are, primarily interested in the 12-month activity level post study? There are 11 regression analyses reported for the secondary outcomes - all at each time point. Is this a useful addition to the manuscript? Who do you want to read this? What message do you want them to receive? I would like some clarification on why there are so many results reported. Alongside all these results are a plethora of p-values which seems unnecessary if you have effect sizes and their precision (see CONSORT, point 17a and 17b).

Response: Thank you for this comment. We agree that the 12-month UCLA score is the primary analysis. This has been sign-posted as such, being the first 'Main analysis' reported (Results, Main analyses, Lines 1-8). Whilst we understand the reviewer's caution regarding the number of analyses reported, to aid transparency of reporting, we have elected to present the whole trial data in an individual publication, rather than splitting data across a number of papers. This has been done to improve reporting, and with the benefit of supplementary files, is hoped not to over-load the reader with data but offer completeness through comprehensive reporting. Accordingly, we feel it is appropriate to report the data as such and have therefore not removed any analyses from this submitted paper. We hope this explanation offers comfort to the reviewer's concern. No confidence intervals have been reported or any formal statistical testing carried out for measures at baseline. The unadjusted mean differences reported for baseline measures are to assess comparability of randomised groups only. Reporting effects at 6-months and confidence intervals was part of the pre-specified analysis plan. It is common in clinical trials to collect and report outcome data at multiple time points, using this longitudinal data in a mixed effects model as done in this trial.

Comment: 10. The Discussion (page 13) is well written and concise. It clearly answers the research question and provides insight into the salient points of the results.

Response: Thank you. No amendment required.

Comment: 11. I recommend that you present relative frequencies in Supplementary Figure 1. On the face of it, it looks like there will be bias introduced by the over representation in the experimental group of various complications. However, the imbalance is due to the imbalance of the groups.

Response: As recommended, this plot has been revised to show proportion of each randomised group experiencing each complication (Supplementary Figure 1).

VERSION 2 – REVIEW

REVIEWER	Hawke, Lyndon J. La Trobe University
REVIEW RETURNED	05-Apr-2022

GENERAL COMMENTS	Thank you for taking on board the reviewers' comments.
--

REVIEWER	Cameron, Claire University of Otago, Dunedin School of Medicine
REVIEW RETURNED	04-Apr-2022

GENERAL COMMENTS	Thank you for the considered response to my questions regarding the original manuscript. I am mostly happy with the revision. I appreciate the greater clarity in the descriptions.
---